# Effect of Family Practice Contract Services on the Perceived Quality of Primary Care among Patients with Multimorbidity: A Cross-Sectional Study in Guangdong, China

**DOI:** 10.3390/ijerph19010157

**Published:** 2021-12-24

**Authors:** Jingyi Liao, Mengping Zhou, Chenwen Zhong, Cuiying Liang, Nan Hu, Li Kuang

**Affiliations:** 1Department of Health Administration, School of Public Health, Sun Yat-sen University, Guangzhou 510080, China; liaojy33@mail2.sysu.edu.cn (J.L.); zhoump@mail2.sysu.edu.cn (M.Z.); 2Jockey Club School of Public Health and Primary Care, The Chinese University of Hong Kong, Hong Kong 999077, China; chenwenzhong@link.cuhk.edu.hk; 3Science Education Department, Dongguan People’s Hospital, Dongguan 523000, China; liangcy3@mail2.sysu.edu.cn; 4Department of Biostatistics, FIU Robert Stempel College of Public Health and Social Work, Miami, FL 33199, USA; 5Department of Family and Preventive Medicine and Population Health Sciences, University of Utah School of Medicine, Salt Lake City, UT 84132, USA

**Keywords:** multimorbidity, family practice contract services, quality of primary care, patient experiences

## Abstract

Family practice contract services, an important primary-care reform policy for improving primary healthcare quality in China, incorporate patients with multiple chronic conditions into the priority coverage groups and focus on their management. This study aims to explore the family practice contract services’ effectiveness in improving the quality of primary care experienced by this population. A cross-sectional study using a three-stage sampling was conducted from January to March 2019 in Guangdong, China. A multivariable linear regression, including interaction terms, was applied to examine the associations between the contract services and primary care quality among people with different chronic conditions. The process quality of primary care was measured in six dimensions using the validated assessment survey of primary care (ASPC) scale. People with contract services scored higher in terms of quality of primary care than those without contract services. Contract services moderated the association between chronic condition status and primary care quality. Significantly positive interactions were observed in the patient-centred care dimension and negative interactions were reflected in the accessibility dimension. Our findings suggest that family practice contract services play a crucial role in improving patient-perceived primary care quality and provide emerging evidence that patients with multimorbidity tend to benefit more from the services, especially in patient-centred care.

## 1. Introduction

The management of patients with multiple, coexisting chronic conditions has been considered a major challenge for governments and health systems worldwide [1,2]. Multimorbidity is considerably associated with high mortality [3], polypharmacy [4], psychological distress [5], reduced quality of life [6] and impaired functional status [7]. Accordingly, patients with multimorbidity become the main users of healthcare, thus increasing the economic burden on the healthcare system [8]. With the prevalence reaching 30% among people aged under 65 years and 55–98% among those aged over 65 years [9,10], multimorbidity becomes more common among older adults, women and those of low socioeconomic status [11]; therefore, the complex care requirements in multimorbidity often do not match the available services [12,13]. This exacerbates the negative impacts on both individuals and the healthcare system, affecting clinical encounters and doctor–patient relationships, and tends to increase clinical complexity and health inequalities [14,15].

The international society has consented that primary healthcare (PHC) is the key response to multimorbidity and awareness of its functional-oriented concept is gradually emerging [16,17]. Generally, the core functions of primary care, namely, first contact, accessibility, continuity, comprehensiveness, coordination and patient-centred care, synergistically constitute a mechanism that promotes health at the primary level and they are key service behaviours in PHC for dealing with multimorbidity [18,19]. Complex interactions in services render patients with multimorbidity more vulnerable to health system failures that potentially result in challenges, such as fragmented services, poorly coordinated and holistic functional care, and costly and potentially dangerous medical errors [20,21,22]. The core function of primary care is largely recognized as helping to address the challenges posed by multiple chronic diseases [23,24].

A growing number of international studies has evaluated the care quality of patients with multimorbidity in terms of the level of achievement of primary care functions [25,26], with studies involving patient assessment indicating that patients with multiple chronic conditions are more likely to report a poorer process quality experience reflected by the primary care core functions than patients with a single chronic condition or no chronic condition. Fung and colleagues [27] found that multimorbidity was associated with lower ratings of communication, although the effects were minor. Gulliford and colleagues [28] reported lower management continuity with more long-term conditions. Paddison and colleagues [29] discovered that patients with multiple long-term conditions less often reported positive patient experience in primary care, when quantified in terms of access, communication and continuity of care. Research evaluating the experiences of patients with multimorbidity from each unique core function of primary care quality is limited. A comprehensive understanding of patient experiences from a multidimensional perspective of primary care is warranted and this would help inform the priorities of service delivery, thus adapting it to the complex needs of patients and improving the quality of their care.

Health systems in developing countries experiencing multi-transition encounter greater challenges in terms of patients with chronic conditions [30]. The prevalence of multimorbidity in China exceeds 10% of the entire Chinese population [17] and 40% of its elderly population [31]; this potentially translates to a substantial increase in the burden on the health system [32]. Established in 2009, China’s new healthcare reform [33] aims to strengthen China’s primary care system. The National Medical Reform Office issued the Guidance on the Promotion of Family Practice Contract Services in 2016 [34] and the family practice contract services’ policy has since become an important part of the primary care reform. The purpose of the contract policy is to fortify basic medical service behaviours and the core characteristic functions of PHC [34]. Family practice contract services are provided in the community health centre mainly by general practitioners, in a care team also composed of nurses and public health physicians. In many countries, including China, general practitioners (GPs) are the main providers of primary care and health services [35,36]. The terms ‘general practitioner’ and ‘family practice’, ‘family doctor’ or ‘family physician’ are used interchangeably in China [37]. Patients voluntarily choose a team of GPs to sign a service contract. The contracts generally last for 1 year and the patients can change the contracted doctors the following year if they are not satisfied with the service [38]. After signing the contract, the physicians provide a free service package for the contracted patients, including creating and managing individual health records, conducting annual health check-ups, implementing proactive life interventions to prevent and manage chronic diseases, follow-ups and referrals. The contracted patients are encouraged to visit their family doctor when they have health problems, but they still reserve the freedom of choosing their preferred medical facilities. The contracted team of GPs is reimbursed on a capitation basis from the public health fund, social insurance pool and government instructions. Given the emerging multimorbidity, the National Health Commission of the People’s Republic of China has classified patients with chronic conditions into the key coverage group in its family doctor contract policy [34].

Previous studies have reported that the family practice contract services’ policy can improve the process quality experience of patients [39,40], but it is not clear whether this policy has an equivalent effect on perceived care quality among patients with multiple chronic diseases, with a single chronic disease and with no chronic diseases and whether patients with multiple chronic diseases can benefit to a greater extent than those among the other two groups.

The purpose of this study is to explore the effect of the family practice contract services’ policy on patients with chronic diseases from the perspective of patients’ process quality experience and provide a basis for subsequent policy improvement. This study addresses three research questions, as follows:Is there a difference in the primary care quality experienced by patients with multiple chronic conditions and by those with a single chronic condition or those without a chronic condition?In the patient groups with varying numbers of chronic conditions, is the perceived quality of patients with contracted services better than those without contracted services? In what ways?Are the family practice contract services more conducive to improving the quality of primary care for patients with multimorbidity than those with single or no chronic condition? What primary care domains do they affect?

## 2. Materials and Methods

### 2.1. Study Design and Participants

We conducted a cross-sectional survey between January and March 2019 in Guangdong Province, southern China. A three-stage cluster sampling method was performed to obtain the samples. In the first stage, 10 cities were sampled in Guangdong Province using a probability-proportional-to-size (PPS) without replacement method considering factors of population composition and primary care resource. In the second stage, 16 healthcare organisations were sampled from the 10 cities with the same probability-proportional-to-size modelling. In the third stage, patients who left the GP’s room were invited to participate in the survey voluntarily by convenience sampling in the waiting area of each healthcare organization. A flowchart of the study participant sampling strategy is shown in Appendix A.

Data were collected using face-to-face and one-to-one interviews. Five postgraduate students with a public health major and advanced training by two researchers waited in the waiting area of the GP’s room at each sampled healthcare organisation and invited patients who left GP’s room to participate in the survey. At the survey site, the investigators did their best to invite every eligible patient to be interviewed. The eligibility criteria included patients who visited the same community health centre (CHC) at least three times, were older than 18 years, could clearly express themselves in either Mandarin or Cantonese and verbally consented to participate in the study. Those who had very poor physical or mental health or had difficulty understanding the questionnaire were excluded. The interviewers presented informed patient consent forms to eligible subjects; explained the purpose of the study completely and carefully; ensured that data were collected using strict principles of anonymity and confidentiality, that is, they ensured that the survey would not have any adverse effect on subsequent visits to the interviewed patients; and obtained informed verbal consent from patients before conducting the interviews. The interviewers would re-check the completeness of the data and would ask the patients regarding the missing questions before ending the survey. As a reward, a small gift (worth about USD 1) was promised to the study participants before the survey and was given to the participant right after the completion of the interview. Refusal to participate or to discontinue participation at any time was allowed. The paper questionnaires and the electronic data obtained from the survey were kept strictly confidential by the supervisor and data administrator. The Institutional Review Board (IRB) of Sun Yat-sen University reviewed and approved the protocol of the study in compliance with the Declaration of Helsinki (Medical Ethics [2018]014).

### 2.2. Measures

#### 2.2.1. Patient Experience Measures

We assessed the patient-perceived primary care process quality using the Assessment Survey of Primary Care (ASPC) scale (See Appendix B), which is a newly developed primary care assessment tool created by our research team in China. Firstly, we constructed a theoretical framework for assessing quality of primary care, which was locally adaptable and internationally comparable. Secondly, we used both qualitative and quantitative research methods to develop and validate items included in the scale, with the aim to ensure that the ASPC can be applied to the local primary care practice and is feasible for social context [41,42]. The ASPC has been confirmed to be a considerably reliable and valid instrument, with an accumulative variance of 62.68% and overall Cronbach’s α of 0.915 [41]. Forty-one items were included in this questionnaire to assess six core dimensions of primary care, namely, first contact, accessibility, continuity, comprehensiveness, coordination and patient-centred care [43]. A 4-point Likert-type scale was used to evaluate the extent to which patient received services (1 = never, 2 = sometimes, 3 = often and 4 = always). Additional options included ‘don’t know/not sure’. The value of which was set at 2.5, in concordance with other similar studies [44,45]. The final score of each dimension was the average score of all items in that dimension. The ‘total ASPC score’ was the average score of all dimension scores on the scale and it was calculated to reflect the overall primary care experience. The scores were converted to a percentage system.

#### 2.2.2. Family Practice Contract Services

Participants were asked, ‘have you contracted with the primary care physician/team?’ This question reported whether they had contracted with a GP. The response options (yes vs. no) were used to define a binary indicator for “family practice contract services” (1 vs. 0).

#### 2.2.3. Chronic Conditions and Multimorbidity

The survey recorded whether a subject had ever been diagnosed with one or more of the seventeen non-communicable chronic diseases (NCDs). The seventeen non-communicable chronic diseases included hypertension, dyslipidaemia, diabetes, cerebrovascular disease, heart diseases, chronic lung diseases, chronic laryngitis, gastroenteritis, peptic ulcer, gallstones and cholecystitis, intervertebral disc disease, rheumatoid arthritis, urolithiasis, prostatic hyperplasia, nephritis and nephropathy, cataract and anaemia. Data were derived from answers to the question, ‘Have you been diagnosed by a doctor with the following 17 NCDs?’. The 17 options presented under the question were subject to classifications used in China’s National Health Service Survey [46].

In this study, multimorbidity was defined as the co-occurrence of two or more chronic conditions in an individual [6,9]. The combination of seventeen measured and reported diseases that were used to count the number of chronic diseases for each participant and those individuals with occurrence of two or more chronic conditions was identified as multimorbidity.

#### 2.2.4. Covariates

The ASPC also includes questions on participants’ socio-demographic characteristics, self-rated health statuses and healthcare service utilization. According to previous studies [9,47], we included the following variables as covariates in the main regression analyses: gender (male or female), age (18–60 or 60 and above), marital status (not married or married), migrant (yes or no), education (primary school or below, middle/high school, or bachelor degree or above), occupation (employed, retired, or unemployed), household income (<5000, 5000–10,000, or >10,000), health status (good, fair, or poor), medical insurance (yes or no) and period of time since the first visit (<2 years, 2~5 years, or >5 years).

### 2.3. Statistical Analyses

Respondents were divided into the following three groups according to the number of chronic conditions: non-chronic condition (NCC) group, single chronic condition (SCC) group and multiple chronic conditions (MCC) group. The NCC group included respondents without non-communicable diseases and the SCC group included those suffering from a single chronic condition. Due to the small sample size (*n* = 187), respondents with two or more noncommunicable diseases (multimorbidity) were merged into one group (the MCC group). Each group was further divided into two subgroups based on whether the patient was contracted to a family practice.

All analyses were conducted using StataCorp (College Station, TX, USA). The purpose of these survey data analyses was to compare the patient-experienced quality of primary care and group differences in the effects of contracted services on the quality of primary care among people with a different number of chronic conditions. Group differences in baseline characteristics and mean scores were evaluated using the chi-squared test, variance analysis and *t*-test. A multivariable linear regression was used to estimate the associations of family practice contract services with primary care attributes in three groups of chronic patients. Further, the interaction term of contract services with chronic conditions was included in the regression model to examine the differences in the relationship of quality scores to the number of conditions between patients who received contract services by a family doctor and those who did not [48,49,50]. The linear multiplicative interaction model was given by the following regression equations:
When the reference group was the non-chronic condition group,
*E* (*Y*) = *µ* + *η**X* + *α*_1_ 1 (*D* = 1) + *α*_1_ 1 (*D* = 2) + *β*_1_ [*X ·* 1 (*D* = 1)] + *β*_2_ [*X ·* 1 (*D* = 2)] + *γ**Z*

where the reference group was the non-chronic condition group, *Y* represents the ASPC scores, *X* represents contract services and *D* represents the category of chronic conditions (*D* = 0, 1, 2 represents the non-chronic condition group, single chronic condition group and multiple chronic conditions group). In addition, [*X ·* 1 (*D* = 1)] and [*X ·* 1 (*D* = 2)] are the interaction terms between contract services and the chronic condition groups; *Z* is the vector of the control variables and *η* indicates the effect of contracted services on the ASPC scores; *α*_1_ and *α*_2_ indicate the effect of chronic disease status on the ASPC scores (single versus non-chronic condition for *α*_1_ and multiple versus non-chronic condition for *α*_2_); *β*_1_ and *β*_2_, respectively, indicate the differences in the effect with and without contract services between single and non-chronic condition and between multiple and non-chronic condition; *µ* represents the constant terms.When the reference group was the single chronic condition group,
*E*(*Y*) = *µ** + *η***X* + *α*_1_* 1 (*D* = 1) + *α*_2_* 1 (*D* = 2) + *β*_1_* [*X ·* 1 (*D* = 1)] + *β*_2_* [*X ·* 1 (*D* = 2)] + *γ*
*Z*
where the reference group was the single chronic condition group, *Y* represents the ASPC scores, *X* represents contract services and *D* represents the category of chronic conditions (*D* = 0, 1, 2 represents the single chronic condition group, non-chronic condition group and multiple chronic conditions group). In addition, [*X ·* 1 (*D* = 1)] and [*X ·* 1 (*D* = 2)] are the interaction terms between contract services and the chronic condition groups; *Z* is the vector of the control variables and *η** indicates the effect of contracted services on the ASPC scores; *α*_1_* and *α*_2_* indicate the effect of chronic disease status on the ASPC scores (non-chronic condition versus single chronic condition for *α*_1_* and multiple versus single chronic condition for *α*_2_*); *β*_1_* and *β*_2_*, respectively, indicate the differences in the effect with and without contract services between non-chronic condition and single chronic condition and between multiple and single chronic condition; *µ* represents the constant terms.

All survey analyses incorporated sampling weights, which were computed to account for differential probabilities of selection due to the nature of the design and to ensure accurate survey estimates [51]. All tests were two-sided and *p*-values < 0.05 were considered statistically significant. Only statistically significant results were interpreted in this work.

## 3. Results

### 3.1. Demographic Characteristics of the Primary-Care Patients

A total of 1185 patients were included in the survey. Of these, 536 participants reported that they had chronic conditions, with 187 of them reporting two or more conditions. Table 1 shows the demographic and health characteristics of respondents who were contracted with a family practice service and those who were not across the three chronic disease groups. Compared with non-contracted respondents, those contracted with a family practice service were predominantly older and retired and tended to have a longer time interval since the first visit to CHC than those who did not.

### 3.2. Patient Primary Care Experience among Individuals with Different Numbers of Chronic Conditions

The individual and total patient primary care experiences are presented overall and by the number of chronic conditions in Table 2. Generally, respondents with multiple chronic conditions reported higher total ASPC scores than those with a single condition (73.11 vs. 71.64, *p* > 0.05) or no chronic condition (73.11 vs. 67.25, *p* < 0.05), although the former was not statistically significant. In comparing the mean score in each of the six primary care dimensions, respondents with multimorbidity reported the highest score in the ‘first contact’ dimension (82.29), followed by ‘patient-centred care’ (79.86); however, their lowest score was in the ‘comprehensiveness’ dimension (61.61). In particular, pairwise comparisons between the MCC and SCC groups, as well as those between the MCC and NCC groups, revealed that the largest score differences were in the ‘continuity dimension’, followed by the ‘first contact’ and ‘patient-centred care’ dimensions.

### 3.3. Relationship between Family Practice Contract Services and Patient Primary Care Experience with Increase in Numbers of Chronic Conditions

Figure 1 exhibits the mean differences and results of regression estimates investigating the relationships between the utilization of family practice contract services and patient primary care experiences with the increase in the number of chronic conditions, after controlling for patient sociodemographic characteristics. Congruent with the results from the bivariate analyses, respondents who contracted with a GP reported a significantly higher total primary care experience score than those who did not (*p* < 0.05), regardless of the number of chronic conditions. In terms of individual attribute scores, respondents with multimorbidity seeing family physicians scored significantly higher in the domains of first contact, continuity, coordination and patient-centred care than non-contracted patients (*p* < 0.05).

Table 3 displays interaction estimates aimed at exploring group differences of the effect of contract services across the three chronic condition groups. Regarding the overall primary care experience, respondents in the MCC group reported an effect difference of 1.55 points higher than those in the SCC group, whereas those in the SCC group reported an effect difference of 1.80 points higher than those in the NCC group, although this was not significant. People with multiple chronic conditions reported a larger effect difference, that is, 3.35 points higher than those without chronic conditions. Specifically, positive interactions across the three groups were observed in the dimensions of ‘first contact’, ‘comprehensiveness’, ‘coordination’ and ‘patient-centred care’ (*β* > 0). In summary, the effect of contract services was greater as the number of chronic conditions increased in the above dimensions and this trend was statistically significantly reflected in the ‘patient-centred dimension’. In other words, family practice contract services interacted with comorbidity status to modify the association with the quality of primary care, which indicates that patients with more chronic diseases benefited more from the contract services. The exception was a negative result across the three groups on the accessibility dimension, although it was non-significant.

The interaction plots illustrate the statistical interaction of contract services with chronic conditions (Figure 2). Corresponding to the results in Table 3, the predicted scores of positive change from 0 to 1 for contracted services in the dimensions of first contact, comprehensiveness, coordination and patient-centred care were the largest in the MCC group and the smallest in the NCC group. However, on the accessibility dimension, there was a negative change in the MCC group, resulting in a negative inter-group comparison of contract services’ effects.

## 4. Discussion

### 4.1. Main Findings

The current study indicates that patients with multiple chronic conditions did not report a worse experience with primary care than those with single or no chronic condition. Family practice contract services were associated with a higher quality of primary care in patients with multimorbidity, among whom a positive effect difference in contract services was reported compared with that among patients with a single chronic condition or no chronic condition. This suggested the contract services moderated the association of comorbidity status and primary care quality to a certain extent. Together, these findings provide additional evidence that having family practice contract services in itself plays a unique role in improving the quality of primary care experienced by patients with many chronic diseases.

### 4.2. Better Patient Primary Care Experience among People with Multimorbidity

Patient experiences have been shown to inform improvements related and complementary to the technical quality of care in health services [52,53]. Our study investigated patient primary care experience among people with multiple conditions in China, based on current knowledge regarding the epidemiology and impact of multimorbidity [4,11]. Unlike our study, previous studies evaluating the experiences of patients with multimorbidity have generally focused on one or certain domains of interest in primary care [29,54,55,56]. Our findings, which resulted from the investigation of patient experiences by evaluating the core features of primary care across six dimensions of care, do not corroborate prior findings regarding the association of multimorbidity with poor care [27,28,29]. In contrast, the current study results demonstrate that a greater number of conditions was associated with higher patient-perceived quality of care. This finding is consistent with that from previous work conducted in the UK [57] and USA [58]. Possible explanations can be traced to specialist involvement and a perceived greater need to provide better care for patients with more conditions. For the former, Takahiro et al. found that the quality score for each additional condition increased more for patients who had seen a relevant specialist than for those who had not [58]. For the latter, Tazeen H et al. concluded that physicians may be more likely to treat hypertension for a patient who has diabetes compared with a patient who does not because of the additional importance of blood pressure control in preserving renal function in diabetic patients [59].

### 4.3. Higher Primary Care Scores for Patients with Multimorbidity Using Family Practice Contract Services

The link between family practice contract services and higher care quality is supported by several other studies [39,60,61]. However, few studies have examined this relationship among multimorbidity patients in terms of the core features of primary care. Our data analysis suggested that the utilization of family practice contract services was associated with a higher quality of care among patients with multimorbidity than patients with a single or no morbidity. This finding does provide support for chronic disease management in the contract services that could improve patients’ perceptions of first contact, continuity, comprehensiveness and coordination care through establishing a stable and long-term doctor–patient relationship.

### 4.4. Effect of Family Practice Contract Services on Primary Care Quality with Increase in Number of Chronic Conditions

Furthermore, we explored the differences in the relationship of quality scores with the number of conditions between patients who received contract services by a family doctor and those who did not. Anecdotally, we found that the contract services interacting with the comorbidity status helped to modify associations with primary care quality. That is, the effect of contract services was greater in several primary care attribute dimensions as the number of chronic conditions increased and significant positive results were obtained on ‘patient-centred care’, except for negative results on accessibility. There are several possible explanations for this finding.

First, as the policy focused more on populations with multimorbidity, family doctors adhering to family practice contract services paid more attention to contract service work for their patients with multiple chronic conditions [62]. These complex patients are rather in need of home service, chronic disease follow-up consultation and other contracted services, compared with those with less complex conditions [57,63]. Therefore, the implementation of family practice contract services plays a modest role in meeting the complex requirements of patients with multimorbidity.

Second, regarding the finding in which a significant trend in the ‘patient-centred care’ dimension was observed, we further compared the items under ‘patient-centred care’ and found that the following items reported significantly larger score differences among people with more conditions: ‘In the process of interacting with the primary care physician/team, did you feel that they cared about you?’ and ‘Did the primary care physician/team explain much to you during the consultation about the cause, condition, diagnosis and medication?’ Additionally, patients with chronic diseases believe that their contracted family doctors are more capable of two-way communication between doctors and patients when making diagnoses and treatment plans and can consciously and actively understand their thoughts [35]. Good communication and active understanding such as this contribute to a better doctor–patient relationship. These findings suggest that contracting GPs can help better understand and respect the views of patients with multimorbidity and are more likely to guide clinical decisions based on the patient’s values instead of pure exposure of the service utilization that is responsible for this association [64], since our results were independent of the time of CHC visits.

Finally, our results do not reflect a positive relationship in the ‘accessibility’ dimension. We subsequently checked the accessibility-related questionnaire items, especially among patients with more than one morbidity, and found that ‘If necessary, could you see the general practitioner on weekends?’ and ‘Did you feel there was a long time to wait outside the consultation room?’ reported significantly lower scores. One possible explanation is that China’s health system does not yet mandate a gatekeeper system for general practitioners and patients remain free to select their doctors, of whom those at higher-level hospitals are preferred. Hence, due to the complexity of patients with multiple conditions, they are more inclined to visit secondary or tertiary hospitals in China in case of emergency [17]. Alternatively, the current number and capacity of family doctors are insufficient to meet the urgent needs of these patients for any emergency health situation occurring at any time [37]. In addition, the lack of an appropriate incentive mechanism for family doctors is also a potential contributor to this result. A previous study showed that there were large differences in financial and non-financial incentives between regions, thus greatly affecting the performance of family doctors [65]. Therefore, without dynamic incentives, family doctors may not be very motivated to provide services to contracted patients outside of working hours. Another explanation is that patients with multimorbidity who have contracted with a family doctor tend to expect a shorter waiting time for a family doctor when they have a health problem; hence, they were more likely to exaggerate their waiting time according to the survey on site.

### 4.5. Study Limitations

This study had some limitations. First, this is a cross-sectional survey that may not have explored the causal relationship between the contract services and the perceived quality experience of patients. Secondly, our research data were exclusively from CHCs within Guangdong Province, thus potentially limiting the extensibility and generalizability of the research results nationwide. Thirdly, our sample of patients with multimorbidity was relatively small, potentially leading to biased results, for which further research needs to be conducted.

## 5. Conclusions

In summary, we found that patients with multimorbidity reported positive experiences in primary care and these patients were more likely to benefit from the contract services provided by family doctors than those with fewer conditions, especially in terms of patient-centred care. The findings of this study suggest that family practice contract services play a crucial role in improving patient-perceived primary care quality and they provide a basis for policymakers to develop the primary healthcare strategies for the management of patients with chronic conditions. The results provide evidence that family practice contract services should focus more on improving the quality of accessibility care for patients with multiple chronic conditions. This requires a more convenient and flexible approach to family doctor visits—an approach designed for the individual, complex needs of patients with multimorbidity—as well as a dynamic and innovative incentive mechanism and continuous quality assurance mechanism for primary care physicians. Further exploration of the relationship between process quality and their clinical outcomes as well as quality of life in multimorbid patients managed by family physicians is imperative.

## Figures and Tables

**Figure 1 ijerph-19-00157-f001:**
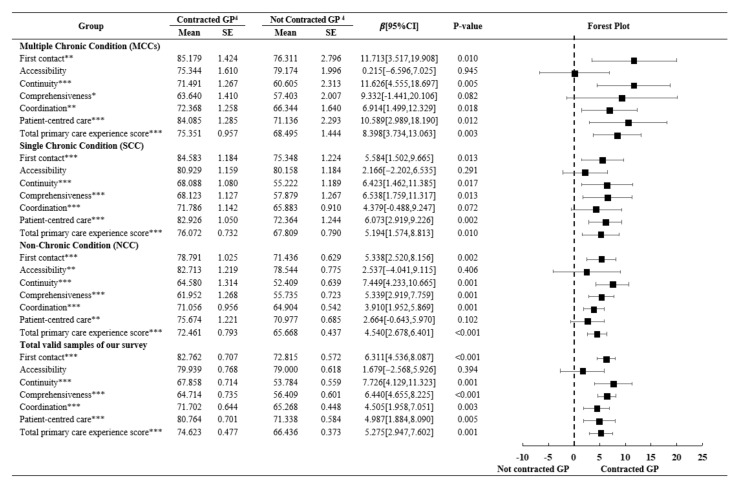
Forest plot of linear regressions of family practice contract services associated with individual and total primary care attribute scores as the number of chronic condition increases. GP, general practitioner. ^4^ Mean and SE of the subgroup with and without contracted GP based on *t*-test. * *p* < 0.05, ** *p* < 0.01 and *** *p* < 0.001, based on *t*-test of difference between those contracted with a GP and those not contracted. The group difference was significant at the 0.05 level, based on multiple regressions results after controlling for gender, age, marital status, migrant, education, occupation, income, health status, medical insurance and period of time since the first visit.

**Figure 2 ijerph-19-00157-f002:**
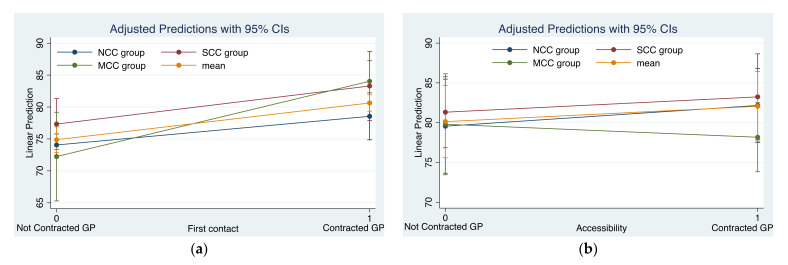
Interaction plots of the adjusted estimates for primary care attribute scores for contract services and chronic conditions. NCC, non-chronic condition; SCC, single chronic condition; MCC, multiple chronic conditions. (**a**) In first contact dimension, the change in predicted scores from not contracted GP to contracted GP for each chronic condition group. (**b**) In accessibility dimension, the change in predicted scores from not contracted GP to contracted GP for each chronic condition group. (**c**) In continuity dimension, the change in predicted scores from not contracted GP to contracted GP for each chronic condition group. (**d**) In comprehensiveness dimension, the change in predicted scores from not contracted GP to contracted GP for each chronic condition group. (**e**) In coordination dimension, the change in predicted scores from not contracted GP to contracted GP for each chronic condition group. (**f**) In patiented-centred dimension, the change in predicted scores from not contracted GP to contracted GP for each chronic condition group.

**Table 1 ijerph-19-00157-t001:** Comparison of demographics and health characteristics of respondents between those contracted with family practice services and those not contracted with family practice services by the number of chronic conditions.

	Multiple Chronic Conditions				Single Chronic Condition				Non-Chronic Condition				
	Contracted with FP, N (%)	Not Contracted with FP, N (%)	*p-*Values ^1^	Total, N (%) ^2^	Contracted with FP, N (%)	Not Contracted with FP, N (%)	*p-*Values ^1^	Total, N (%) ^2^	Contracted with FP, N (%)	Not Contracted with FP, N (%)	*p-*Values ^1^	Total, N (%) ^2^	*p-*Values ^2^
Sample size	126 (67.4)	61 (32.6)		187 (15.8)	162 (46.4)	187 (53.6)		349 (29.5)	151 (23.3)	498 (76.7)		649 (54.8)	
Gender			0.257				0.229				<0.05		<0.001
Male	50 (39.7)	19 (31.1)		69 (36.9)	71 (43.8)	94 (50.3)		165 (47.3)	35 (23.2)	182 (36.5)		217 (33.4)	
Female	76 (60.3)	42 (68.9)		118 (63.1)	91 (56.2)	93 (49.7)		184 (52.7)	116 (76.8)	316 (63.5)		432 (66.6)	
Age			0.172				<0.001				<0.05		<0.001
18~60	24 (19.0)	17 (27.9)		41 (21.9)	72 (44.4)	119 (63.6)		191 (54.7)	115 (76.2)	433 (86.9)		548 (84.4)	
>60	102 (81.0)	44 (72.1)		146 (78.1)	90 (55.6)	68 (36.4)		158 (45.3)	36 (23.8)	65 (13.1)		101 (15.6)	
Marital status			0.205				<0.05				<0.05		<0.001
Not married	1 (0.8)	2 (3.3)		3 (1.6)	1 (0.6)	10 (5.3)		11 (3.2)	7 (4.6)	60 (12.0)		67 (10.3)	
Married	125 (99.2)	59 (96.7)		184 (98.4)	161 (99.4)	177 (94.7)		338 (96.8)	144 (95.4)	438 (88.0)		582 (89.7)	
Migrant			0.127				<0.001				<0.001		<0.001
Yes	16 (12.7)	13 (21.3)		29 (15.5)	43 (26.5)	87 (46.5)		130 (37.2)	45 (29.8)	299 (60.0)		344 (53.0)	
No	110 (87.3)	48 (78.7)		158 (84.5)	119 (73.5)	100 (53.5)		219 (62.8)	106 (70.2)	199 (40.0)		305 (47.0)	
Education			<0.05				0.666				<0.05		<0.001
Primary school or below	42 (33.3)	36 (59.0)		78 (41.7)	48 (29.6)	59 (31.6)		107 (30.7)	45 (29.8)	84 (16.9)		129 (19.9)	
Middle/high school	71 (56.3)	18 (29.5)		89 (47.6)	93 (57.4)	99 (52.9)		192 (55.0)	69 (45.7)	295 (59.2)		364 (56.1)	
Bachelor’s degree or above	13 (10.3)	7 (11.5)		20 (10.7)	21 (13.0)	29 (15.5)		50 (14.3)	37 (24.5)	119 (23.9)		156 (24.0)	
Occupation			<0.001				<0.001				0.102		<0.001
Employed	14 (11.1)	9 (14.8)		23 (12.3)	48 (29.6)	95 (50.8)		143 (41.0)	85 (56.3)	327 (65.7)		412 (63.5)	
Retired	93 (73.8)	25 (41.0)		118 (63.1)	76 (46.9)	43 (23.0)		119 (34.1)	15 (9.9)	43 (8.6)		58 (8.9)	
Unemployed	19 (15.1)	27 (44.3)		46 (24.6)	38 (23.5)	49 (26.2)		87 (24.9)	51 (33.8)	128 (25.7)		179 (27.6)	
Household income (CNY/month)			0.302				0.246				0.065		<0.001
<5000	104 (82.5)	50 (82.0)		154 (82.4)	138 (85.2)	149 (79.7)		287 (82.2)	115 (76.2)	332 (66.7)		447 (68.9)	
5000–10,000	16 (12.7)	5 (8.2)		21 (11.2)	17 (10.5)	22 (11.8)		39 (11.2)	18 (11.9)	96 (19.3)		114 (17.6)	
>10,000	6 (4.8)	6 (9.8)		12 (6.4)	7 (4.3)	16 (8.6)		23 (6.6)	18 (11.9)	70 (14.1)		88 (13.6)	
Health status			<0.05				0.903				0.305		<0.001
Good	28 (22.2)	15 (24.6)		43 (23.0)	48 (29.6)	54 (28.9)		102 (29.2)	73 (48.3)	271 (54.4)		344 (53.0)	
Fair	70 (55.6)	22 (36.1)		92 (49.2)	90 (55.6)	102 (54.5)		192 (55.0)	63 (41.7)	192 (38.6)		255 (30.3)	
Poor	28 (22.2)	24 (39.3)		52 (27.8)	24 (14.8)	31 (16.6)		55 (15.8)	15 (9.9)	35 (7.0)		50 (7.7)	
Medical insurance			0.763				<0.001				<0.001		<0.001
Yes	121 (96.0)	58 (95.1)		179 (95.7)	159 (98.1)	161 (86.1)		320 (91.7)	144 (95.4)	412 (82.7)		556 (85.7)	
No	5 (4.0)	3 (4.9)		8 (4.3)	3 (1.9)	26 (13.9)		29 (8.3)	7 (4.6)	86 (17.3)		93 (14.3)	
Period of time since the first visit			0.074				0.167				<0.001		<0.001
<2 year	9 (7.1)	11 (18.0)		20 (10.7)	23 (14.2)	39 (20.9)		62 (17.8)	20 (13.2)	171 (34.3)		191 (29.4)	
2~5 year	30 (23.8)	14 (23.0)		44 (23.5)	36 (22.2)	46 (24.6)		82 (23.5)	37 (24.5)	123 (24.7)		160 (24.7)	
>5year	87 (69.0)	36 (59.0)		123 (65.8)	103 (63.6)	102 (54.5)		205 (58.7)	94 (62.3)	204 (41.0)		298 (45.9)	

FP, family practice. ^1^
*p*-values are based on χ^2^ tests of differences between those contracted with an FP and those not contracted in the three chronic condition groups, respectively. ^2^ Total number of persons in the group and the percentage of the group in the total respondent and *p*-values are based on χ^2^ tests of differences among the three groups.

**Table 2 ijerph-19-00157-t002:** Individual and total primary care attribute scores reported by respondents by the number of chronic conditions.

	MCC Group, Mean (SE)	SCC Group, Mean (SE)	NCC Group, Mean (SE)	F-Values	*p*-Values ^1^	Mean Difference		
						MCC Group Versus SCC Group	MCC Group Versus NCC Group	SCC Group Versus NCC Group
First contact	82.29 (1.35)	79.63 (0.89)	73.15 (0.55)	34.86	<0.001	2.65	9.14 *	6.49 *
Accessibility	76.59 (1.27)	80.52 (0.83)	79.51 (0.66)	3.49	<0.05	−3.92 *	−2.92	1.00
Continuity	67.94 (1.20)	61.19 (0.88)	55.24 (0.61)	50.67	<0.001	6.75 *	12.70 *	5.95 *
Comprehensiveness	61.61 (1.17)	62.63 (0.90)	57.18 (0.64)	14.49	<0.001	−1.03	4.42 *	5.45 *
Coordination	70.40 (1.02)	68.62 (0.74)	66.34 (0.48)	8.49	<0.001	1.78	4.07 *	2.29 *
Patient-centred care	79.86 (1.22)	77.27 (0.87)	72.07 (0.60)	23.41	<0.001	2.59	7.79 *	5.20 *
Total primary care experience score	73.11 (0.83)	71.64 (0.59)	67.25 (0.40)	31.77	<0.001	1.47	5.87 *	4.40 *

^1^ Significance indicated at *p* < 0.05, based on variance analysis of differences among different number of chronic groups. * The mean difference was significant at the 0.05 level, based on multiple comparisons of differences in different numbers of chronic groups using the Bonferroni test.

**Table 3 ijerph-19-00157-t003:** Linear regression of family practice contract services on primary care attribute scores among the three different chronic condition groups, including interactions.

	First Contact, *β* (95%CI)	Accessibility, *β* (95%CI)	Continuity, *β* (95%CI)	Comprehensiveness, *β* (95%CI)	Coordination, *β* (95%CI)	Patient-Centred Care, *β* (95%CI)	Total Primary Care Experience Score, *β* (95%CI)
The reference group is the non-chronic condition group (NCC)							
Contracted GP × Group (ref.: non-chronic condition)							
Contracted GP × Group (SCC)	1.48 [−4.10, 7.06]	−0.67 [−8.14, 6.79]	−0.53 [−5.63, 4.57]	3.58 [−2.17, 9.34]	1.53 [−4.68, 7.73]	5.41 [0.89, 9.92] *	1.80 [−2.42, 6.03]
Contracted GP × Group (MCC)	7.32[ −2.39, 17.02]	−4.23 [−11.97, 3.50]	2.94 [−4.50, 10.37]	4.17 [−7.47, 15.80]	2.22 [−3.92, 8.36]	7.68 [1.22, 14.14] *	3.35 [−1.86, 8.56]
The reference group is the single chronic condition group (SCC)							
Contracted GP × Group (ref.: single chronic condition)							
Contracted GP × Group (NCC)	−1.48 [−7.06, 4.10]	0.67 [−6.79, 8.14]	0.53 [−4.57, 5.63]	−3.58 [−9.34, 2.17]	−1.53 [−7.73, 4.68]	−5.41 [−9.92, −0.89] *	−1.80 [−6.03, 2.43]
Contracted GP × Group (MCC)	5.84 [−2.28, 13.95]	−3.56 [−10.71, 3.59]	3.47 [−5.04, 11.98]	0.58 [−12.25, 13.42]	0.69 [−5.51, 6.90]	2.27 [−5.90, 10.45]	1.55 [−2.80, 5.90]

* The group difference was significant at the 0.05 level, based on multiple regressions results after controlling for gender, age, marital status, migrant, education, occupation, income, health status, medical insurance and period of time since the first visit.

## Data Availability

The data presented in this study are available on request from the corresponding author.

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
