# Peer review of "Effect of Family Practice Contract Services on the Perceived Quality of Primary Care among Patients with Multimorbidity: A Cross-Sectional Study in Guangdong, China"

_ijerph, 2021, doi:10.3390/ijerph19010157_

Round 1

Reviewer 1 Report

Great article that provides great results, I do wish that in the conclusions you could represent in a more complete way the results obtained in your article.

Author Response

Comments and Suggestions for Authors

Great article that provides great results, I do wish that in the conclusions you could represent in a more complete way the results obtained in your article.

Responses to comments and explanation of the revisions:

       Thank you for your valuable suggestion. We have added crucial findings of the study, their research implications, and future research directions in the conclusion of the manuscript. The specific contents are as follows:

        In summary, we found that patients with multimorbidity reported positive experiences in primary care, and these patients were more likely to benefit from the family practice contract services than those with fewer conditions, especially in patient-centred care. The findings of this study suggest that family practice contract services play a crucial role in improving patient-perceived primary-care quality and provide a basis for policymakers to develop the primary health-care strategies for the management of patients with chronic conditions. The results provide insight that the family practice contract services should focus more on improving the quality of accessibility care for patients with multiple chronic conditions. This requires a more convenient and flexible approach to family doctor visits designed for the individual and complex needs of patients with multimorbidity, as well as a dynamic and innovative incentive mechanism and a continuous quality assurance mechanism for primary care physicians. Further exploration of the relationship between process quality and their clinical outcomes as well as quality of life in multimorbid patients managed by family physicians is imperative. (Lines 546-556 Page 17).

        We have also uploaded the above response as an attachment.

Reviewer 2 Report

This is a very interesting and well written article. The approach employed by researches is robust, the size of empirical sample also. I have only minor indications for improvement:

  1. Please shorten the abstract. It is longer than 200 words recommended by IJERPH.
  2. Conclusion section is too general and provides very limited value to readers. There are lots of papers on perceived quality of primary care among patients. Such general presentation provides no added value. Please select from discussion section the key findings which the authors consider are worth being provided as final conclusions. 

Author Response

Comments and Suggestions for Authors

      This is a very interesting and well written article. The approach employed by researches is robust, the size of empirical sample also. I have only minor indications for improvement:

  1. Please shorten the abstract. It is longer than 200 words recommended by IJERPH.

Responses to comments and explanation of the revisions:

     We thank you for your valuable comments and feedback. We have shortened the abstract to less than 200 words. (Lines 16-29, Page 1). We selected the primary study objectives and findings and have streamlined the presentation of the study methods. After introducing the complete name of the family practice contract services, we abbreviated it as contract services to meet the word count requirement. Further, we have ensured that the abstract is concise, without missing any crucial aspect of the study.

  1. Conclusion section is too general and provides very limited value to readers. There are lots of papers on perceived quality of primary care among patients. Such general presentation provides no added value. Please select from discussion section the key findings which the authors consider are worth being provided as final conclusions.

Responses to comments and explanation of the revisions:

      Many thanks for your valuable suggestions. We have added crucial findings of the study, their research implications, and future research directions in the conclusion of the manuscript. The specific contents are as follows:

       In summary, we found that patients with multimorbidity reported positive experiences in primary care, and these patients were more likely to benefit from the family practice contract services than those with fewer conditions, especially in patient-centred care. The findings of this study suggest that family practice contract services play a crucial role in improving patient-perceived primary-care quality and provide a basis for policymakers to develop the primary health-care strategies for the management of patients with chronic conditions. The results provide insight that the family practice contract services should focus more on improving the quality of accessibility care for patients with multiple chronic conditions. This requires a more convenient and flexible approach to family doctor visits designed for the individual and complex needs of patients with multimorbidity, as well as a dynamic and innovative incentive mechanism and a continuous quality assurance mechanism for primary care physicians. Further exploration of the relationship between process quality and their clinical outcomes as well as quality of life in multimorbid patients managed by family physicians is imperative. (Lines 546-556 Page 17).

We have also uploaded the above response as an attachment.

Reviewer 3 Report

...After reading the whole article once, I started commenting on the introduction and methods. After the "methods" I stopped because I did not see essential prerequisites for "good scientific practice" fulfilled.

There is no written consent from the patients. I did not find any information concerning data management and/or data security or data protection, and details of patient recruitment are unclear.

Author Response

Comments and Suggestions for Authors

   ...After reading the whole article once, I started commenting on the introduction and methods. After the "methods" I stopped because I did not see essential prerequisites for "good scientific practice" fulfilled.

     There is no written consent from the patients. I did not find any information concerning data management and/or data security or data protection, and details of patient recruitment are unclear.

Responses to comments and explanation of the revisions:

    Thanks for your valuable comments, and we apologize for any confusion and the lack of clarity. This is a cross-sectional study conducted from January to March 2019 in Guangdong, China. The study was conducted according to the guidelines of the Declaration of Helsinki, approved by the Institutional Review Board of Sun Yat-sen University (Medical Ethics [2018]014), and funded by the National Natural Science Foundation of China, grant number 71673311. We have also added this declaration to the revised manuscript. (Lines 180-182, Page 4; Lines 564-565, Page 17).

     Before the interview, our investigators clarified the contents of the informed consent form to the patient, ensured the anonymity and confidentiality of the data collected, that the survey situation would not have any adverse effect on subsequent visits to the interviewed patients, and obtained verbal consent from each patient. We have added data management and protection and patient recruitment details in the method section of the revised manuscript. (Lines 156-180, Pages 3-4). We have also refined the presentation of the regression equations in the statistical analysis section. (Lines 263-352, Pages 5-6). Additionally, we have structured our article according to the STROBE guidelines for cross-sectional studies and tried our best to improve the manuscript.

      We have also uploaded the above response as an attachment.

Round 2

Reviewer 3 Report

I habe ethical concerns, which I wrote to the editors.

Author Response

Responses: Thanks for your valuable comments, and we apologize for any confusion and the lack of clarity. To response the ethical concerns, we want to respond and provide detailed information regarding the important issue.

1. Our study protocol and practice strictly followed the guidelines of the Declaration of Helsinki and was approved by the Institutional Review Board of Sun Yat-sen University (“the IRB” hereafter). The study investigators paid extreme attention to protect our survey participants during the entire study process from data collection to data analysis and interpretations. We never collected any information of the study participants that could be used to identify their identities, including (but not limited to) their name, national ID number, driver’s license number, MRN, and their residential/mailing addresses. In addition, the survey interviewers were also trained to not ask for any information regarding their interviewee’s identification.  

2. We did not add any additional items in survey questionnaires that were approved by the IRB. In reality, at the end of each interview our survey interviewer would always re-check the completeness of the questionnaire data. If any of the item in the questionnaire was incomplete, the corresponding questions would be asked again, and the answers would be added to the missing items in case the study participants gave his/her answers at that time [1-3]. In the previous version of the manuscript, we mis-stated "added the missing items" that may cause a confusion about the process. This should be corrected as "the interviewers would re-check the completeness of the data and would ask the patients regarding the missing questions before ending the survey". We have modified the presentation in the manuscript accordingly. Please see below or see lines 151–153, page 3 in the manuscript.

“The interviewers would re-check the completeness of the data and would ask the patients regarding the missing questions before ending the survey.” (Lines 151-153, Page 3)

3. In China, it is typical to use the “on-spot” survey method to conduct survey, instead of using the mail-in or IT based surveys.[4-26]

4. It is common to give a small gift, as our incentive, to survey participants to promote and the participation and to show our appreciation to the participants. The gifts usually worth 5-10 Yuan RMB (about 1 USD). The gifts were promised before conducting the survey and given to the participants right after the end of the survey. It was NOT sent to the address of the participants. In addition, no contact information was collected in order to them the incentives. Refusal to participate or to discontinue participation at any time was allowed. We made a checklist of literature on the same type of studies (Attachment 1) [1-3,7,27-36]. In the previous version of our manuscript, we inappropriately addressed this as "a small present was sent to patients". We have corrected this to “A small gift (worth about $1 USD) was promised to the study participants before the survey and was given to the participant right after the completion of the interview”. Please see below or see lines 153–156, page 3 in the manuscript.

“As a reward, a small gift (worth about $1 USD) was promised to the study participants before the survey and was given to the participant right after the completion of the interview. Refusal to participate or to discontinue participation at any time was allowed.” (Lines 153-156, Page 3)

Further, we have used an English-language editing service to address the language issues.

We have also uploaded the above response including  references and Certificate of editing as an attachment.
